# Estimating Total Length of Partially Submerged Crocodylians from Drone Imagery

Clément Aubert [1,2,3,4,5,*], Gilles Le Moguédec [2], Alvaro Velasco [6], Xander Combrink [7], Jeffrey W. Lang [8], Phoebe Griffith [9], Gualberto Pacheco-Sierra [10], Etiam Pérez [11], Pierre Charruau [12], Francisco Villamarín [13], Igor J. Roberto [14], Boris Marioni [15], Joseph E. Colbert [16], Asghar Mobaraki [17], Allan R. Woodward [18], Ruchira Somaweera [19,20], Marisa Tellez [21], Matthew Brien [22] and Matthew H. Shirley [1,4]

1   Global Forensics and Justice Center, Florida International University, 8285 Bryan Dairy Rd. #125, Largo, FL 33777, USA; mshirley@fiu.edu
2   AMAP, University Montpellier, CIRAD, CNRS, INRAE, IRD, 34000 Montpellier, France; gilles.moguedec@cirad.fr
3   Institut des Sciences de l'Evolution de Montpellier (UMR 5554, CNRS-UM-IRD-EPHE), Université' de Montpellier, Cedex 5, 34095 Montpellier, France
4   Project Mecistops, 615 Waterside Way, Sarasota, FL 34242, USA
5   Nature Conversv'Action, 28 Avenue Georges Clémenceau, 34430 Saint Jean de Védas, France
6   FUDECI (Foundation of the National Academy of Physical, Mathematical and Natural Sciences), Av. Universidad, Bolsa a San Francisco, Edif, Palacio de las Academias, Edif Anexo, Piso 2, Of Fudeci, Distrito Capital, Caracas 1010-A, Venezuela; velascocaiman@gmail.com
7   Department of Nature Conservation, Tshwane University of Technology, Pretoria 0001, South Africa; combrinkas@tut.ac.za
8   Gharial Ecology Project, Madras Crocodile Bank Trust, East Coast Road, Vadanamelli Village, Mahabalipuram 603104, Tamil Nadu, India; jeff.lang@email.und.edu
9   Leibniz Institute of Freshwater Ecology and Inland Fisheries, Müggelseedamm 310, 12587 Berlin, Germany; phoebe.griffith@igb-berlin.de
10  Be'Tonal Conservation & Research AC, Calle 5n x30 y 32 #405, Mérida 97236, Yucatán, Mexico; gpacheco@iecologia.unam.mx
11  Enterprise for the Conservation of the Zapata Swamp, km 30 Jaguey Grande-Playa Larga Street, Zapata Swamp 43000, Cuba; cocodrilo@eficz.co.cu
12  Departamento de Conservación de la Biodiversidad, El Colegio de la Frontera Sur, Unidad Villahermosa, Carretera Villahermosa-Reforma km 15.5, Ranchería el Guineo Sección 2, Villahermosa C.P. 86280, Tabasco, Mexico; pierre.charruau@ecosur.mx
13  Grupo de Biogeografía y Ecología Espacial (BioGeoE2), Universidad Regional Amazónica Ikiam, km 7 vía Muyuna, Tena 150101, Ecuador; francisco.villamarin@ikiam.edu.ec
14  Faculdade de Educação, Ciências e Letras de Iguatu, Universidade Estadual do Ceará, Iguatu 63500-000, Ceará, Brazil; igor.joventino@uece.br
15  Instituto Nacional de Pesquisa da Amazonia INPA, Programa PCI, Av. Andre Araujo, Manaus 6900-000, Amazonas, Brazil; borispcca@gmail.com
16  Jekyll Island Conservation Department, 100 James Rd., Jekyll Island, GA 31527, USA; jcolbert@jekyllisland.com
17  Wildlife Conservation and Management Bureau, Department of Environmental, Pardisan Eco-Park, Tehran P.O. Box 14155-7383, Iran; a.mobaraki@doe.ir
18  Florida Fish and Wildlife Research Institute, Florida Fish and Wildlife Conservation Commission, 1105 SW Williston Road, Gainesville, FL 32601, USA; allan.woodward@myfwc.com
19  Stantec Australia, Perth, WA 6000, Australia; ruchira.somaweera@stantec.com
20  School of Environmental and Conservation Sciences, Murdoch University, Murdoch, WA 6150, Australia
21  Crocodile Research Coalition, Maya Beach, Stann Creek, Belize; marisa.tellez@crcbelize.org
22  Queensland Parks and Wildlife Services, Department of Environment and Science, Manly, QLD 4179, Australia; matt.brien@des.qld.gov.au
*   Correspondence: clement.aubert34@hotmail.fr; Tel.: +33-6-59-85-41-89

**Abstract:** Understanding the demographic structure is vital for wildlife research and conservation. For crocodylians, accurately estimating total length and demographic class usually necessitates close observation or capture, often of partially immersed individuals, leading to potential imprecision and risk. Drone technology offers a bias-free, safer alternative for classification. We evaluated the effectiveness of drone photos combined with head length allometric relationships to estimate total

length, and propose a standardized method for drone-based crocodylian demographic classification. We evaluated error sources related to drone flight parameters using standardized targets. An allometric framework correlating head to total length for 17 crocodylian species was developed, incorporating confidence intervals to account for imprecision sources (e.g., allometric accuracy, head inclination, observer bias, terrain variability). This method was applied to wild crocodylians through drone photography. Target measurements from drone imagery, across various resolutions and sizes, were consistent with their actual dimensions. Terrain effects were less impactful than Ground-Sample Distance (GSD) errors from photogrammetric software. The allometric framework predicted lengths within ≃11–18% accuracy across species, with natural allometric variation among individuals explaining much of this range. Compared to traditional methods that can be subjective and risky, our drone-based approach is objective, efficient, fast, cheap, non-invasive, and safe. Nonetheless, further refinements are needed to extend survey times and better include smaller size classes.

**Keywords:** UAV; allometry; crocodiles survey; non-invasive survey; ecology; alternative methods

## 1. Introduction

Drones are now established and widely used as useful tools for conservation science and natural resource management [1,2]. They have the ability to collect very high-resolution (e.g., 4K and 8K) images [3], perform long-range autonomous flights to collect data in inaccessible areas [4,5], and are much less costly than helicopters and small manned planes [6,7]. Moreover, they are easy to pilot and generally only require a short training [8], and most newer brands result in limited disturbance to wildlife due to noise suppression techniques [9,10]. Constant improvements to battery life, sensors, and embedded algorithms continuously increase the capacity and utility of drones. In particular, the developing diversity of on-board cameras, including higher-resolution or multispectral cameras, LiDAR systems, allows for new means to remotely investigate ecosystems in ever increasing detail and spectrums. Drones have already been used to evaluate the behavior [11], population ecology [12–14], anti-poaching [15,16], and habitat monitoring [5] related to a diversity of species. However, standardized and objective testing is still needed to validate specific sampling approaches, particularly in respect to specific species, contexts, and environments [2].

In particular, drones have high potential for the study of cryptic, shy, and dangerous species, as well as those that inhabit places difficult to access. This is the case of crocodylians—which are largely cryptic, mostly nocturnal, and mostly aquatic. In areas where multiple species coexists, survey methods ideal for one species may not necessarily suit the others [17]. Because of their unique natural histories, crocodylians pose many challenges for researchers and wildlife managers seeking to implement efficient monitoring protocols. Surveys from a boat, on foot, or from manned aircraft are the traditional methods used for crocodylians [18–21]; they are widely used and proven effective [22,23]. Nonetheless, they are time-consuming, and often require significant, thus costly, human resources and heavy logistics. In comparison, during a comparable amount of time, a single drone-operator could cover a comparable amount of habitat without incurring additional field-related expenses or risk to personnel. Traditional on-ground methods are further limited by often inaccessible wetlands and other habitats that are difficult to navigate and penetrate, which are not barriers for drones. However, drones also have their drawbacks: they are limited by battery life and the logistics required to recharge batteries or have ample backups in the field; they are limited by available memory to store large quantities of data (e.g., photos and videos); they are limited to observations in the open with no visibility of individuals under vegetation; they are largely restricted to daytime operations with very limited or no capacity to have sufficient lighting sources; and there are increasing administrative and legal restrictions on their usage in some regions.

Drones have recently been used in several crocodilian studies, mostly to count the number of individuals or nests, but also for individual identification, including for the American alligator *Alligator mississippiensis* [24,25], American crocodile *Crocodylus acutus* [26], Yacare caiman *Caiman yacare*, and broad-snouted caiman *Caiman latirostris* [27], saltwater crocodile *Crocodylus porosus* [28–31], mugger crocodile *Crocodylus palustris* [32,33], Nile crocodile *Crocodylus niloticus* [34–36], West African crocodile *Crocodylus suchus* [10], and gharial *Gavialis gangeticus* [37]. Aubert et al. (2021) proposed optimal drone flight parameters to count crocodiles under various field conditions and compared their efficiency to other survey methods. Beyond population size surveys, biometric data that could be obtained during the surveys can provide a greater understanding of demographic and reproductive variables, leading to better understanding and managing the populations [38]. In general, crocodilian surveys do attempt to collect data on individual body size and size-class structure of populations [39,40].

Collection of such data for crocodylians, however, is not easy. Precise body measurements for population size structure requires capturing and restraining individuals, which can be logistically intense and requires trained personnel to ensure human and crocodile safety [41]. It additionally causes a high amount of stress on the captured individuals, potentially even leading to death from anoxic acidosis and other factors [42–46]. Alternatively, size can be estimated visually where observers approach the specimen as closely as possible to identify the species and evaluate its total length (TL), but this relies heavily on observer experience, can be subjective from a distance, and can be heavily biased [47]. Most of the time, however, only the head is visible in partially submerged individuals and total length (TL) is estimated from head length (HL) using personal knowledge, experience, or formulas [48], thus leading to further bias [49].

Drones provide a potential solution to most of these challenges because they are capable of capturing standardized, high-resolution photos from which the size of an object could be remotely measured. Based on our knowledge, two previous studies have attempted to measure size of crocodiles using drones [33,34]. However, neither evaluated the accuracy of their measurements, therefore the biases by ground topology, camera definition, software, and observer, the crocodile size or body position is unknown. Moreover, these studies focused only on fully visible (i.e., unobstructed) crocodiles on land, thus limiting the number of individuals measured relative to the number detectable in photos.

In this study, we seek to validate a protocol and algorithm for using aerial drone photos to estimate accurately and precisely the TL of individual crocodylians from their HL. We first assessed the accuracy of measurements for different covariables and flight or photo parameters (e.g., topology, target size, camera definition, software, observer, biological variations). Then, using previous capture records of real HL:TL ratios for 17 crocodylian species, we defined a method for drone-based individual size estimation. Finally, we tested the application of this method using available photo data of *Crocodylus suchus* from previous drone crocodile surveys [10].

## 2. Materials and Methods

### 2.1. Study Areas and Equipment

We used MAVIC 2 Pro (DJI, Shenzhen, China) operated from a Samsung Galaxy Tab 6 (Samsung, Seoul, Republic of Korea) to measure standard targets placed in an open and flat field area outside of Montpellier, France. This grey colored drone has a maximum flight time ~31 min, a maximum speed of 72 km per hour, a pilot-controlled range of $\pm 6$ km in optimal conditions, a sound of $\approx 75$ dB, and weighs 907 g. It was equipped with a high-definition camera L1D-20c (4K/30 fps; 20 MP 1″ sensor; Hasselblad, Gothenburg, Sweden). The images using for the study were collected by Aubert et al. (2021) in 2017 and 2018 from several ponds in the Pendjari National Park (Benin) and transects along the Tapoa River in the W National Park (Niger).

## 2.2. Calibration of Flight and Photo Parameters for Optimal Measurements

We first estimated how photo resolution impacts the precision of measurements using standardized targets. We made four targets of each size to mimic the approximate length and volume of crocodile heads of three different sizes (Size 1 = 13.7 cm; Size 2 = 27.3 cm; Size 3 = 40.8 cm). We flew the drone over the targets at different altitudes (20, 30, and 40 m) corresponding to different photo resolutions (0.51, 0.76 and 1.02 cm$^2$ per pixel), following the optimal flight parameters as recommended by [10] (60% overlap, flight speed 5 m·s$^{-1}$, 90° camera orientation, autonomously following a pre-programmed flight plan from take-off to landing). We programmed all flight plans using the Pix4D capture software version 4.5.4 (Pix4D, Prilly, Switzerland). We replicated flights at each altitude six times, resulting in 18 maps covering ca. 0.7 ha each. During the flights, we recorded the time of the day and visually scored the cloud cover from 0 to 4 (0 = 0%, 1 = 1–25%, 2 = 26–50%, 3 = 51–75% and 4 = 76–100%). We used Agisoft Metashape Pro (ver. 1.6.2.10247, Agisoft, St Petersburg, Russia) to assemble and ortho-rectify all images, and imported them into QGIS (ver. 3.16.16, QGis Development Team, USA) for analysis.

Once pre-programmed, consumer drones fly at a constant altitude, irrespective of the variations in topology along the flight path (e.g., slopes, depressions, etc.), therefore could have varying ground resolutions in the captured images. Following recommendation issued from [10], we privileged the 40 m altitude. To assess the impact of the small elevation differences one would expect in a natural wetland system on measurement accuracy, we conducted an additional three flights at each of five different altitudes (38, 39, 40, 41 and 42 m) within the same test area. By comparing the resolution at these different altitudes, we thus mimic the ±2 m variation in topology that we would expect flying along a river course. Indeed, some crocodiles are in the water and some are on the river banks at varying points in the slope. For each flight, we calculated the Ground-Sample Distance (GSD, in cm/pixel) using two independent methods: (1) automatically, using the Agisoft Metashape Pro software (thus the GSD as estimated by the software), and (2) using the formula in Equation (1) [50], thus the actual GSD.

$$GSD_h = \frac{Flight\ height \times Sensor\ height}{Focal\ length \times Image\ height} \quad (1)$$

For each generated image, we measured the target lengths using the "Measure Line" tool in QGIS, which requires the user to manually define two reference points on the picture corresponding to the extremities of the target; then the length of the defined segment is converted from pixels to centimeters using the resolution. To estimate the repeatability, the same user measured each target twice from the same photo within a 3-week interval.

## 2.3. Allometric Ratios for Total Length Determination

We sought to establish the allometric relationship between head length (HL, as measured from tip of snout to dorsal supraoccipital margin) and total body length (TL, as measured ventrally from tip of snout to tip of tail) in order to create a reference allometric framework that incorporates sources of variation relevant to measuring HL and estimating TL from drone photos. To do this, we compiled HL and TL measured from captured individuals of 17 different crocodylian species: *Alligator mississippiensis* (USA (Florida): 1650, USA (Georgia): 744), *Caiman crocodilus* (Brazil: 459), *Crocodylus acutus* (Belize: 259, Mexico (Atlantic): 493, Mexico (Pacific): 154), *Crocodylus intermedius* (Venezuela: 403), *Crocodylus johnstoni* (Australia: 588), *Crocodylus moreletii* (Belize: 390, Mexico: 207), *Crocodylus niloticus* (Egypt: 65, Gabon: 20, South Africa: 228, Tanzania: 5, Uganda: 22), *Crocodylus palustris* (India: 23, Iran: 57), *Crocodylus porosus* (Australia: 370), *Crocodylus rhombifer* (Cuba: 196), *Crocodylus suchus* (Côte d'Ivoire: 24, Ghana: 37, Niger: 36, Uganda: 19), *Gavialis gangeticus* (India: 308, Nepal: 45), *Mecistops leptorhynchus* (Gabon: 159), *Melanosuchus niger* (Brazil: 104, Ecuador: 63), *Osteolaemus tetraspis* (Gabon: 106), *Paleosuchus palpebrosus* (Brazil: 149), and *Paleosuchus trigonatus* (Brazil: 87). The final database included HL and TL measurements from 7368 individual crocodylians from these species (Table 1).

**Table 1.** Summary statistics for the 17 Crocodylians species and summary of the two modeling approaches (ratio and allometry). For each crocodilian species, we provide the number of measured individuals ($N_i$), the number of observations kept for modeling ($N_u$, see text), as well as mean (Mean) and median (Med) of head and total lengths (HL and TL) with variation on each measurement across the sample (1st quartile Q1 and 3rd quartile Q3). We also provide statistics information for the two methods to estimate TL from HL: (1) for the ratio approach: observed 1st quartile (Q1), median (Med), mean (Mean) and 3rd quartile (Q3) of TL:HL ratio; (2) for the allometry, Log TL = a + b * Log HL + $\varepsilon$: estimated coefficients (a, b), residual standard deviation of $\varepsilon$ ($\sigma$) and the $R^2$ (determination coefficient) of the regression, as well as the relative error (RE) as a percentage. The contribution of each source to the total imprecision is also indicated: head inclination (HI), head length measurement (HLM), allometry variation (AV) and allometry residuals (AR).

| Species | $N_i$ | $N_u$ | HL (cm) | | | | TL (cm) | | | | | | Ratio TL/HL | | | | Allometry Characteristics | | | | RE | Variance Distribution by Sources of Imprecision | | | |
|---|---|---|---|---|---|---|---|---|---|---|---|---|---|---|---|---|---|---|---|---|---|---|---|---|---|
| | | | Q1 | Med | Mean | Q3 | Min | Q1 | Med | Mean | Q3 | Max | Q1 | Med | Mean | Q3 | a | b | $\sigma$ | $R^2$ | (%) | HI (%) | HLM (%) | AV (%) | AR (%) |
| *Alligator mississippiensis* | 2391 | 2374 | 13.5 | 17.5 | 19.1 | 25.5 | 20.6 | 99.9 | 130.9 | 140.8 | 189.1 | 396.0 | 7.1 | 7.4 | 7.3 | 7.6 | 1.89 | 1.04 | 0.06 | 0.99 | 10.90 | 2.5 | 43.9 | 2.7 | 50.9 |
| *Caiman crocodilus* | 459 | 454 | 14.6 | 18.0 | 16.6 | 19.3 | 26.4 | 109.2 | 132.3 | 122.3 | 143.5 | 204.9 | 7.1 | 7.4 | 7.3 | 7.6 | 1.81 | 1.06 | 0.06 | 0.97 | 12.20 | 2.3 | 40.0 | 2.5 | 55.3 |
| *Crocodylus acutus* | 906 | 905 | 4.2 | 7.8 | 12.6 | 18.6 | 22.5 | 27.1 | 49.0 | 82.4 | 121.4 | 372.0 | 6.3 | 6.4 | 6.4 | 6.6 | 1.82 | 1.02 | 0.05 | 1.00 | 9.70 | 2.7 | 48.2 | 3.0 | 46.1 |
| *Crocodylus intermedius* | 403 | 396 | 5.7 | 7.3 | 8.3 | 9.3 | 23.7 | 35.2 | 46.8 | 51.2 | 56.5 | 197.0 | 6.0 | 6.1 | 6.3 | 6.4 | 1.84 | 1.00 | 0.08 | 0.96 | 14.80 | 1.6 | 28.5 | 1.6 | 68.2 |
| *Crocodylus johnstoni* | 588 | 539 | 4.0 | 4.6 | 7.2 | 9.6 | 18.6 | 25.9 | 29.4 | 42.4 | 55.0 | 230.2 | 6.1 | 6.4 | 6.4 | 6.7 | 2.02 | 0.91 | 0.06 | 0.98 | 13.00 | 1.7 | 29.9 | 1.5 | 67.0 |
| *Crocodylus moreletii* | 597 | 591 | 8.1 | 12.6 | 15.2 | 20.9 | 21.0 | 53.0 | 85.9 | 102.3 | 139.5 | 375.0 | 6.4 | 6.7 | 6.7 | 6.9 | 1.83 | 1.03 | 0.06 | 0.99 | 12.50 | 2.1 | 37.5 | 2.4 | 58.0 |
| *Crocodylus niloticus* | 340 | 340 | 4.2 | 9.6 | 19.0 | 38.7 | 27.2 | 32.0 | 71.2 | 136.6 | 275.1 | 413.6 | 7.1 | 7.3 | 7.3 | 7.6 | 2.01 | 0.99 | 0.07 | 1.00 | 13.50 | 1.8 | 32.3 | 1.9 | 64.0 |
| *Crocodylus palustris* | 80 | 79 | 21.0 | 31.0 | 34.3 | 47.8 | 43.0 | 144.5 | 196.5 | 206.5 | 260.3 | 487.0 | 5.5 | 6.5 | 6.4 | 7.2 | 2.49 | 0.81 | 0.12 | 0.94 | 24.30 | 0.5 | 8.9 | 0.4 | 90.2 |
| *Crocodylus porosus* | 370 | 368 | 6.4 | 8.2 | 10.5 | 11.5 | 26.9 | 41.5 | 54.3 | 71.3 | 77.9 | 332.5 | 6.5 | 6.6 | 6.7 | 6.8 | 1.78 | 1.05 | 0.04 | 0.99 | 8.40 | 3.2 | 55.7 | 3.3 | 37.9 |
| *Crocodylus rhombifer* | 196 | 193 | 15.5 | 22.8 | 25.0 | 34.0 | 94.0 | 109.8 | 163.0 | 176.5 | 234.0 | 330.0 | 6.9 | 7.1 | 7.1 | 7.2 | 2.05 | 0.97 | 0.05 | 0.98 | 10.10 | 2.5 | 43.7 | 2.2 | 51.6 |
| *Crocodylus suchus* | 116 | 115 | 7.0 | 10.3 | 13.9 | 18.4 | 34.2 | 49.8 | 69.1 | 94.8 | 126.0 | 250.0 | 6.7 | 6.9 | 6.9 | 7.1 | 2.02 | 0.96 | 0.05 | 0.99 | 9.40 | 2.7 | 46.9 | 2.7 | 47.8 |
| *Gavialis gangeticus* | 353 | 350 | 30.0 | 40.0 | 41.1 | 52.3 | 73.0 | 172.0 | 223.0 | 230.7 | 293.0 | 533.0 | 5.4 | 5.6 | 5.6 | 5.9 | 1.76 | 0.99 | 0.08 | 0.95 | 15.10 | 1.6 | 27.9 | 1.5 | 69.1 |
| *Mecistops leptorhynchus* | 159 | 159 | 8.6 | 12.2 | 17.0 | 21.1 | 33.8 | 50.7 | 70.1 | 96.1 | 120.4 | 302.0 | 5.6 | 5.8 | 5.7 | 5.9 | 1.83 | 0.97 | 0.03 | 1.00 | 5.80 | 3.8 | 66.6 | 3.4 | 26.1 |
| *Melanosuchus niger* | 167 | 167 | 9.2 | 15.1 | 17.1 | 23.8 | 31.2 | 73.5 | 121.1 | 131.9 | 188.7 | 283.5 | 7.5 | 7.7 | 7.7 | 8.0 | 2.02 | 1.01 | 0.06 | 0.99 | 11.20 | 2.4 | 41.0 | 2.4 | 54.2 |
| *Osteolaemus tetraspis* | 106 | 103 | 9.1 | 12.1 | 13.0 | 17.1 | 39.5 | 61.3 | 81.8 | 88.9 | 112.9 | 165.2 | 6.6 | 6.8 | 6.7 | 6.9 | 1.83 | 1.03 | 0.05 | 0.98 | 10.30 | 2.6 | 45.0 | 2.8 | 49.6 |
| *Paleosuchus palpebrosus* | 149 | 148 | 8.0 | 11.9 | 12.4 | 15.9 | 28.1 | 54.2 | 84.1 | 88.2 | 120.1 | 185.5 | 6.9 | 7.1 | 7.1 | 7.3 | 1.80 | 1.07 | 0.05 | 0.99 | 9.50 | 2.9 | 50.4 | 3.3 | 43.5 |
| *Paleosuchus trigonatus* | 87 | 87 | 12.9 | 15.7 | 16.2 | 19.5 | 50.0 | 81.3 | 102.8 | 103.8 | 127.7 | 183.0 | 6.2 | 6.4 | 6.4 | 6.6 | 1.65 | 1.08 | 0.04 | 0.98 | 7.70 | 3.4 | 58.5 | 3.3 | 34.8 |

We calculated the allometric ratio for each individual, and ultimately removed all observations for which the allometric ratio was 1.4 > HL:TL > 1.9 as these likely represent measurement or data recording errors (i.e., they do not correspond to real crocodiles). To better understand the precision of our allometric ratios and the estimated TL from HL for realistic crocodile lengths, we simulated the entire process with random perturbations at each step. This allowed us to provide confidence intervals around the predicted values that consider all known sources of imprecision. These sources of imprecision were either directly measured from our experiments or estimated by simulating perturbations, as follows:

- *Head inclination*: because the drone camera objective is vertically oriented, direct estimation of the HL from the picture implicitly assumes that the head is horizontally oriented. In reality, head inclination can deviate from the horizontal plane due to the terrain slope, because crocodylians thermoregulate by opening their mouths, or when they are simply resting at any non-horizontal angle. This leads to an underestimation of the real HL by a $cos(\theta)$ factor, where $\theta$ is the angle between the head inclination and the horizontal plane (Figure 1). We simulated head inclination using a $\beta$ distribution for $\theta \in [0°; 90°]$ (Figure S1a). Since we have no data to fit that inclination, we arbitrarily chose the distribution parameters so that the average inclination $\bar{\theta}$ equals 5° and that $\theta < 20°$ for 99% of the samples, a conservative choice.

- *Target length estimation:* we compared the lengths measured in drone photos lengths ($HL_e$) to the know lengths ($HL_0$) of the mock targets. The imprecision of the $HL_e$ measurement ($\varepsilon_h$) can result from variation of the distance between the ground and the drone altitude (due to topology), orthophoto treatment, or observer accuracy in choosing the two reference points (i.e., head delimitation effect; Figure 1). We measured this imprecision as $\varepsilon_h = ln\frac{HL_e}{HL_0}$ and fitted a Johnson's SU-distribution, a 4-parameter distribution that is more flexible than the classical normal distribution. In particular, this distribution can be asymmetric. We used the logarithm of the relative error, rather than the absolute error, to stabilize the variance (heteroscedasticity). We tried both Gaussian and Johnson's SU distributions, where goodness-of-fit indicated that Johnson's SU better fit the data (Figure S1b).

- *Allometry*: to take into account the natural variability of individuals and the limited size of the sample of allometry data above, we used a simple linear regression *ln(TL) = f[ln(HL)]* to predict TL from HL, and to estimate the confidence interval around TL for a given HL. The logarithm is used to stabilize the residual variance, in accordance with the standard hypothesis of the linear model. Overall, the total imprecision on the total body length prediction ($TL_e$) is thus the consequence of all these independent sources of imprecision (head inclination, target length acquisition, and allometry). We simulated them 50 times each to produce the overall confidence intervals around $TL_e$, thereby establishing a robust reference allometric framework. We then determined the part of the total deviance of $TL_e$ from TL explained by each source using an ANOVA. We performed all analyses in R version 4.2.2 [51].

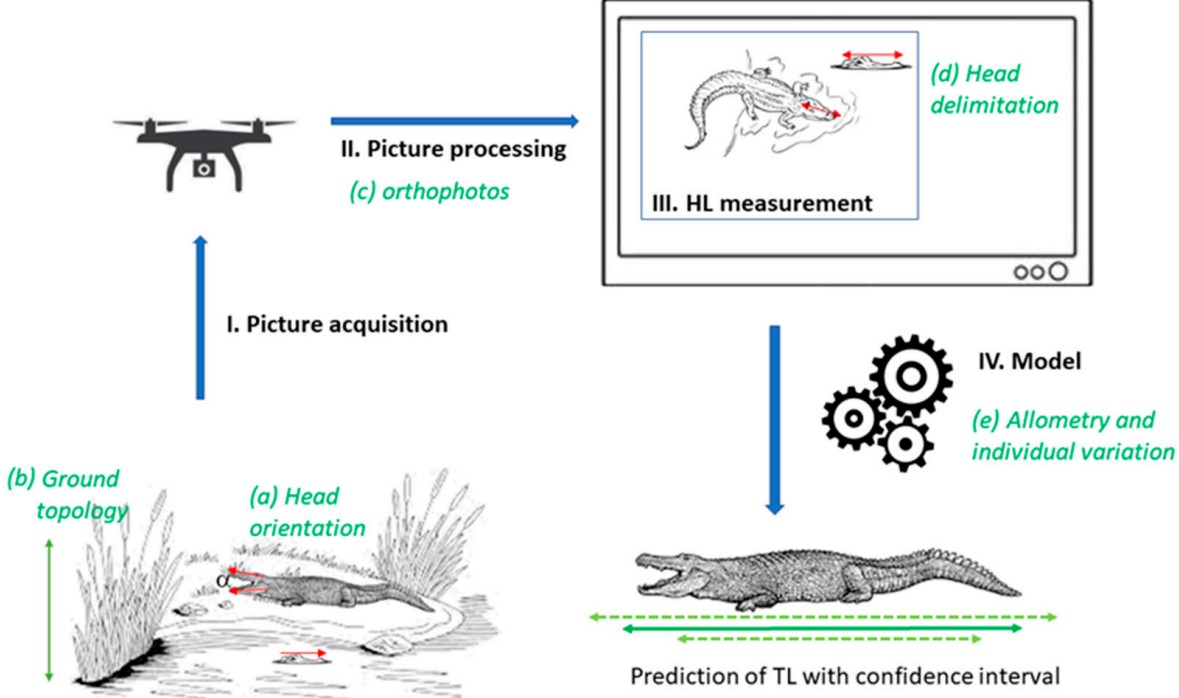

**Figure 1.** Estimating the total length of crocodylians from drone-captured images and the various sources of imprecision. This schematic represents the various steps (bolded) to obtain demographic information from crocodiles observed by drone: I. picture capture with the drone; II. image processing; III. measurement of head length (HL); and IV. model-based estimation of the total body length (TL) of the crocodile (dashed lines represent the CI on the TL estimation). The various sources of imprecision at each step are also indicated (green, italicized): (*a*) head inclination, (*b*) topology, (*c*) orthophoto treatment (i.e., software effects), (*d*) head delimitation (i.e., observer effects), and (*e*) allometry and individual variation.

### 2.4. Crocodile Size Class Distribution in Natural Populations

We used drone images of *Crocodylus suchus*, previously collected in 2018 in the Tapoa River (W National Park, Niger) and Bali Pond (Pendjari National Park, Benin) [10], to test our reference allometric framework. For all individuals appearing in full view in the photos, we measured both $HL_p$ and $TL_p$, i.e., the head and body lengths measured directly from the drone-captured and ortho-rectified photos, with QGIS (Figure 2). Using our reference allometric framework designed for *C. suchus*, we then estimated $TL_e$ from the measured $HL_p$ and compared it back to $TL_p$ for each individual. Some crocodiles may have been photographed multiple times, but because each photo represents different acquisition conditions (e.g., time of the day, ambient light, body position, etc.) we considered that each crocodile in each photo represents a unique sample point for testing the reference allometric framework. For estimating the size class distributions, for sites where multiple flights were flown, we only used the single flight that detected the highest number of crocodiles.

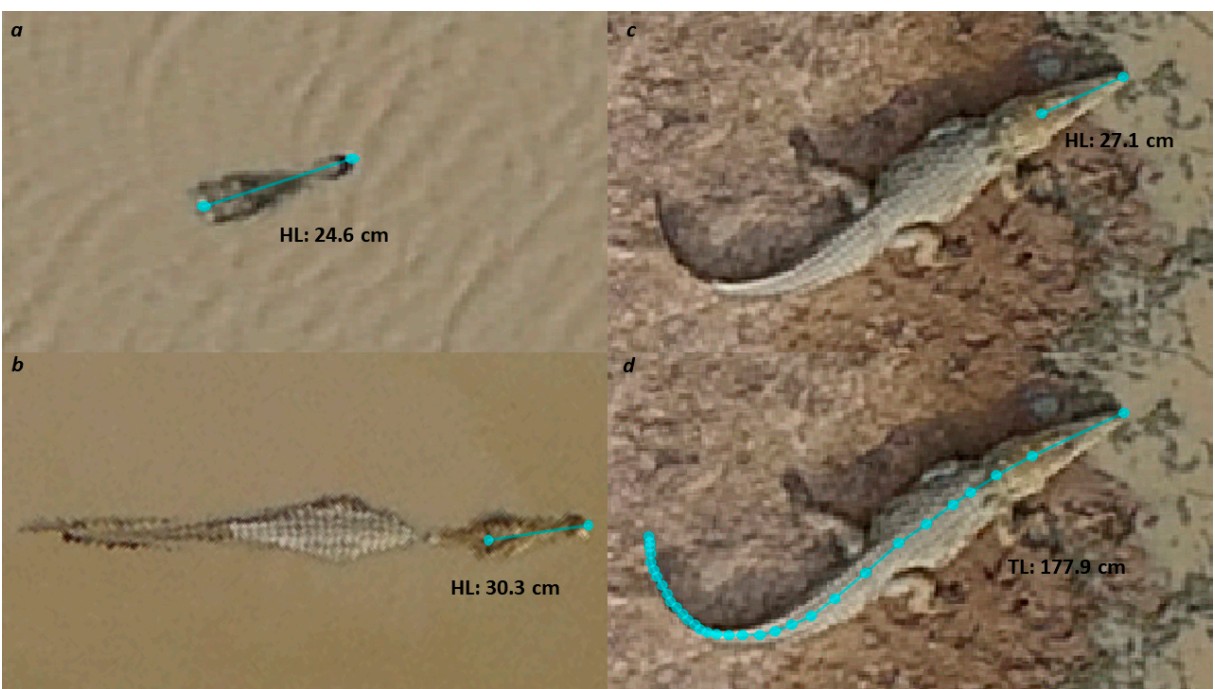

**Figure 2.** *Crocodylus suchus* measurements using QGIS. For head length (HLp): (**a**) partially submerged with only the head visible, (**b**) partially submerged, (**c**) on the shore. For total length (TLp): (**d**) on the shore. Pendjari National Park (Benin) and W National Park (Niger).

## 3. Results and Discussion

The aim of the present study was to develop a robust, remote, and objective method for the collection of biometric data from photographed crocodylians as part of drone crocodile surveys, in particular the total length (TL) of observed individuals. Such data are indeed invaluable to understand crocodylian demographics for both research and management purposes [18,38].

### 3.1. Drone-Captured Pictures Allow Precise Target Length Measurement

We generally found that the drone-based measurements are very accurate. Flight height (20, 30 and 40 m) resulting in slight differences of image resolution (0.51, 0.76 and 1.02 $cm^2$ per pixel) had no significant effect on the precision of the measurement, which remains on the order of $\pm 1$ cm (Table 2, Figure 2a). Similarly, the standard deviation of the difference was around 0.8 cm for all target sizes (Table 2, Figure 3b). There was no apparent effect of cloud cover or time of day. This precision of the measurement is largely enough for our purpose. In particular, the 40 m flight altitude, and its corresponding image resolution of 1.02 $cm^2$ per pixel, recommended for drone surveys of crocodiles [10] appears well adapted. Note that, as part of our previous work [10], we showed that *C. suchus* was not affected by drones at flight altitudes > 10 m, though [31] observed that *C. porosus* responded to drones at 30 m. We are not aware of any other studies on disturbances to crocodylians due to drones, and it is not yet known whether different drones with different power and/or sound profiles will have different impacts, nor could we presume to know the results of such tests for other crocodylian species. Although there is already some literature on the impact of drones on wildlife (birds: [3,9,52,53], mammals: [54–56], and reptiles and fish: [57]), disturbance tests should be performed before implementing any drone survey.

However, our results suggest several additional considerations. First, we found a clear bias in that TL tended to be systematically, although marginally (median difference $\approx$ 0.5 cm), overestimated (>75% of the measurements, Figure 3). This is probably mostly due to the pixelated nature of the image. At the tested image resolutions, the target size

was quite small, making it difficult to decide between adjacent pixels when delimiting the end points. We tended to include the last pixel, which partly surpasses the end of the target. This may be related to a second source of imprecision, which is observer heterogeneity in choosing the measurement endpoints. From one measurement to the next three weeks later, we found a mean difference of ±0.43 cm (95% of the differences were between −1.09 cm and +1.80 cm) and a standard deviation of the variability of 0.76 cm (Figure S1b).

**Table 2.** Description of targets used to calibrate the length measurement methodology. For each target true length and flight altitude at which the targets were photographed and measured, we provide the average measurements of targets, standard deviation of targets, difference between average measurement and true target length (Δ), and the relative error of the measurement (as a percent deviation from the true value).

| True Length (cm) | Altitude (m) | Average Estimation (cm) | Standard Deviation (cm) | Δ (cm) | Relative Error (%) |
|---|---|---|---|---|---|
| 13.7 | 20 | 14.02 | 0.58 | 0.32 | 2.4 |
| 13.7 | 30 | 14.30 | 0.82 | 0.60 | 4.4 |
| 13.7 | 40 | 14.22 | 0.72 | 0.52 | 3.8 |
| 27.3 | 20 | 27.70 | 0.71 | 0.40 | 1.4 |
| 27.3 | 30 | 27.99 | 0.92 | 0.69 | 2.5 |
| 27.3 | 40 | 27.95 | 0.82 | 0.65 | 2.4 |
| 40.8 | 20 | 41.09 | 0.51 | 0.29 | 0.7 |
| 40.8 | 30 | 41.75 | 1.10 | 0.95 | 2.3 |
| 40.8 | 40 | 41.80 | 0.67 | 1.00 | 2.5 |

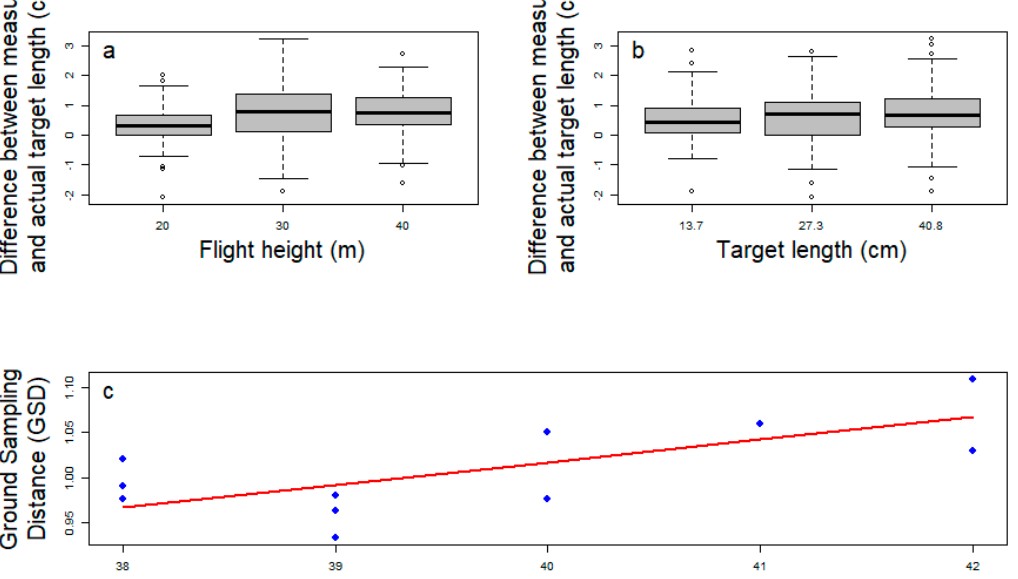

**Figure 3.** Effect of flight height, target length and ground topology on measurement precision in drone photos. Parts (**a,b**): Distribution of the difference between target length as measured in the drone photo and actual target length at different flight heights (**a**) and for different target lengths (**b**). The boxplots represent the median, 25% and 75% quartiles, whiskers representing 5% and 95% quartiles, and dots the outliers. Part (**c**): the true Ground-Sample Distance (GSD) was calculated relative to flight height (red line, see Equation (1)) and compared to the GSD automatically estimated by the photogrammetric processing software Agisoft Metashape Pro (blue dots) to assess whether minor differences (±2 m) in ground topology would affect the estimation. The inaccuracy automatically computed (±0.070 cm/pixel at 40 m) was larger than the true difference in GSD (0.025 cm/pixel per meter). The topology effect is thus overshadowed by the GSD variation in the processing software.

Finally, within a single photo or in reconstructed orthophotos, two targets could actually be at different distances from the drone camera due to micro-topological variation (e.g., crocodiles can rest on rocks or fallen trees, etc., resulting in uneven ground). However, we did not observe any significant effects. For example, when GSD was estimated automatically by Agisoft Metashape Pro, the inaccuracy ($\pm$0.070 cm/pixel at 40 m) was larger than the true difference in GSD (0.025 cm/pixel per meter; hence $-$0.05 cm/pixel at 38 m and +0.05 cm/pixel at 42 m) (Figure 3c). The topology effect is thus overshadowed by the GSD variation in the processing software and as long as the difference in level does not exceed $\pm$2.8 m, then the error will be limited to 0.025 cm/pixel per meter. To maintain a constant altitude during the flight, elevation information can be used by Shuttle Radar Topography Mission (SRTM) to establish an automated flight plan wherein the drone maintains a constant altitude relative to terrain heterogeneity along the flight path, but this makes set-up more complex and more costly for virtually no benefit.

This level of imprecision (<1 cm; Table 2, Figure 3a) has very little, if any, impact on how these data are ultimately used in crocodylian monitoring programs. Crocodylians are usually categorized into size classes based on either broader life stage categories or estimated total length categories. The first method comprises subjective categories such as hatchlings (youngs of the year), juveniles, subadults, or adults, which are often defined by some ecological criteria (e.g., probability to disperse or ability to reproduce), differ from one species to another, and between males and females within a species [58–60]. The second method is based on visual estimation of total length and typically the observed animals are grouped into 25 or 50 cm classes [61–64]. In both cases, the groupings of individuals are quite coarse with length thresholds that far surpass 1 cm variation. Moreso, biologically, within and between species, sexes and geographical locations, there is notable variation in the size thresholds. Thus, the minor imprecision in measurements in drone-based methods will have no biologically meaningful effect on the outcomes.

### 3.2. Reference Allometric Framework for Estimating Total Length from Head Length in Crocodylians

Two prior studies included size class estimations of individuals from drone surveys, but only measured individuals fully visible on the banks [33,34]. Most crocodylians, however, spend most of their time at least partially submerged with only their heads visible [49]. Providing a population-level demographic classification based only on crocodiles basking on land would inevitably make the assessment biased towards larger size classes because smaller crocodiles are less likely to be fully emerged and visible on banks [10,65,66]. Thus, the ability to estimate the total body length (TL) of an individual crocodile from its head-length (HL) provides an alternative to measure the size of individuals that are partially submerged. For many crocodylian species, the typical HL:TL allometric relationship is about at 1:7 ratio [48,67], but the exact ratio differs between crocodylian species [48]. Moreover, even within the same species, the allometric ratio can also vary with the size, particularly for very large animals [68–71]. For example, the ratio of very large *C. porosus* seems to be closer to 1:8 [49]. Drone-captured images allow more accurate and repeatable measurements, thus, providing a great advantage over the traditional on-ground visual estimation approach, though still acknowledging that between and within-species variation may bias the estimated TL from HL measurements in the photos.

The simple HL:TL ratio calculated for the 17 different species varied from 1:5.6 to 1:7.3 (Table 1). These results are similar to previous studies estimating this relationship for some of these species [48,49,72–76], while for others this is the first time this information is being published (e.g., *C. suchus*, *M. leptorhynchus*). With the exception of *C. palustris*, in 16 of the 17 species, the slope of the allometric method was also very close to 1 in logarithmic scale reinforcing the usefulness of the ratio for both these species and for our sample (Table 1). If the allometry slope were strictly equal to 1, the two methods (ratio and allometric) would lead to the same estimation, but only the allometric method provides information on bias and precision (Figure 4). Deviation from 1 in the allometric method predicts an ontogenetic shift in

the relationship with crocodile size (and thus age), where slope of <1 predicts that TL increases less than proportionally with HL, and >1 predicts that it increases more than proportionally (as observed by [77]). For *C. porosus*, we found a slope equal to 1.05 possibly driven by the HL:TL ratio which is closer to 1:8 (than 1:7 as in other species) for the larger individuals. In the allometric method, the intercept coefficients were quite variable between species (Table 1, Figure 4), again reinforcing the interspecific differences. The species for which the relationship was the least accurate was *C. palustris* (Figure S9), which is likely due to the small sample size for this species that came from two different populations with different size class distributions and for which the method of measurement was unknown. It should be emphasized that the residual analysis for each of the 17 regressions showed no particular problems, and that the R-squared are very close to 1. The worst of these is for *C. palustris*, equal to 0.94 (Table 1).

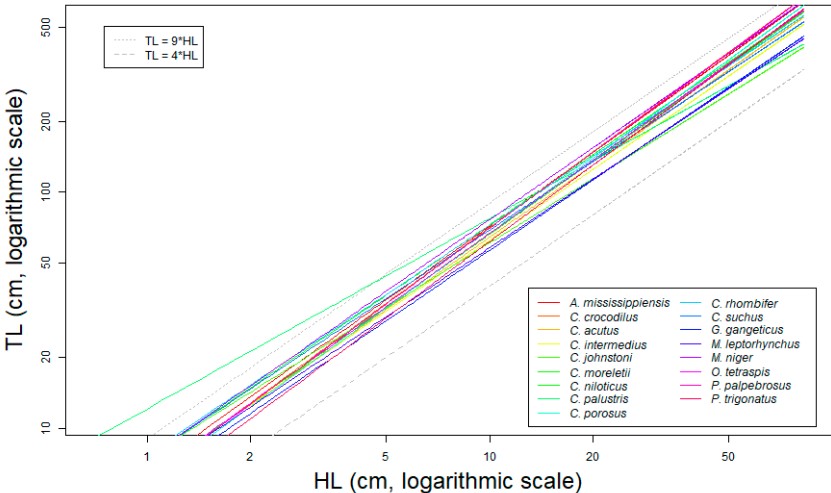

**Figure 4.** Simple allometric relationship between head length (HL) and total length (TL) in log–log scale for 17 crocodylian species. The allometric relationships were derived from measurement of 7368 wild-caught individuals (Table 1). For the allometry computation, only data considered as realistic, those with the TL:HL ratio between 4 and 9, have been used.

The allometry corresponds to one step of the whole process to estimate TL. Each step of the HL acquisition process and of its conversion to TL indeed contains its own sources of bias or imprecision, which we treated as follows:

- *Measurement bias*: We accounted for the measurement imprecision in drone photos previously identified from the standard targets by using a Johnson's SU-distribution, which better fit the data than a Gaussian distribution (Figure S1b). The Johnson's SU-distribution was fitted on the logarithm of the relative measurement error and the value of its four parameters are: gamma = 0.0947, delta = 0.936, xi = 0.0209 and lambda = 0.0227.
- *Head inclination:* The drone objective is perpendicular to the ground, thus if the target is not horizontal its size can be underestimated (see Methods, Figure 1). This could be particularly problematic to measure crocodile head length because crocodylians often incline their head. We assessed this potential distortion by conservatively assuming that, on average, crocodiles have a head inclination of 5° and 99% of the population have a head inclination < 20° (Figure S1a; pers. obs.). With this assumption, we calculated that we underestimate the true length in drone photos by 0.7% on average, and that the underestimate is less than 6% for 99% of the population. Randomly adding target inclination distortions in our model further confirmed that it results in limited relative imprecision (2.7% of total variability).
- *Allometric variation*: For all species, we observed a robust allometric relationship between HL and TL (See Table 1, Figures 5a and S2a–S17a). Our data show that the absolute variation of the allometric relationship increases with the size of the individuals (i.e., more

variance around the predicted values for bigger crocodylians), but with a fairly constant relative error (average $\simeq$ 9.63%, range $\simeq$5.8% for *M. leptorhynchus* to $\simeq$15% for *G. gangeticus*, not including *C. palustris*, which we excluded because of the afore-explained data quality problems). As it was directly measured on real crocodiles, this variation comes from biological processes independent from the measuring method.

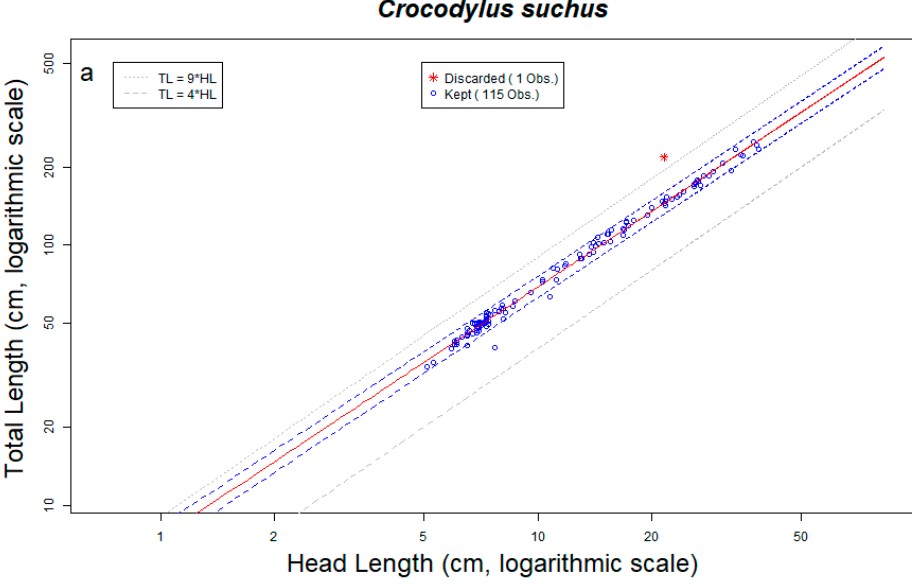

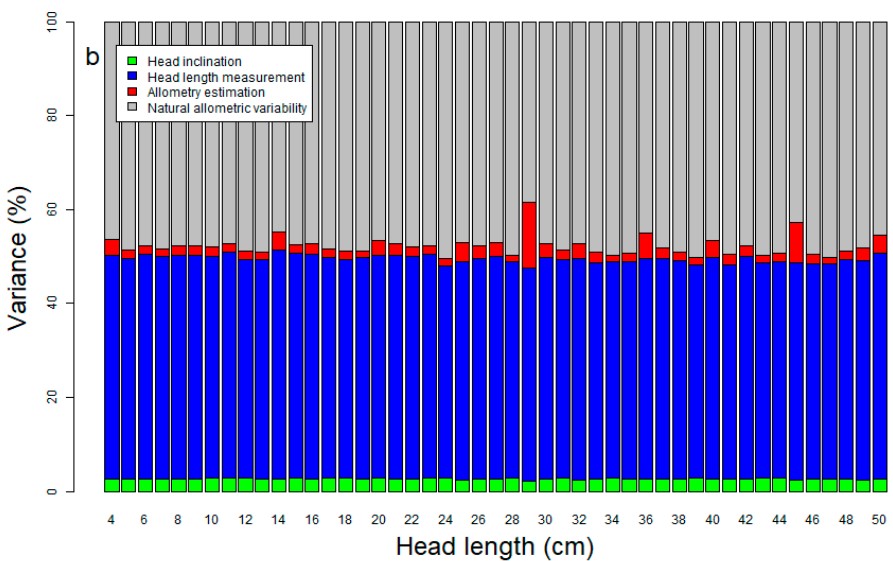

**Figure 5.** Allometric relationship between head length (HL) and total length (TL) in log–log scale of wild-caught West African crocodiles (*Crocodylus suchus*) measured in natural populations. (**a**) The allometric relationship is derived from HL and TL measurements from 116 individual *C. suchus* captured from throughout the species distribution. We discarded all observations (red stars; kept individuals are indicated by a blue dot) for which the ratio was greater than 1:9 (grey dotted line) and less than 1:4 (grey dashed line). The allometry prediction curve (red line) and its 95% confidence envelope (blue lines) are illustrated. (**b**) We estimated the variance by simulating 125,000 values (i.e., 50 head inclinations × 50 target length acquisitions × 50 allometry values randomly chosen from their respective distributions) to assess the contribution of each source of bias to the overall imprecision in the predicted total length estimations based on the allometric relationship: (i) head inclination (light green), (ii) head length measurement (blue), (iii) allometry estimation (red), and (iv) natural allometric variability, i.e., biological variation (grey).

Overall, despite several sources of imprecision, we were able to design a reference allometric framework based on a statistical model to estimate TL from HL with a robust confidence interval for 17 different crocodylian species (See Figures 5a and S2a–S17a, Tables S1–S17). The advantage of our method is that it offers an objective estimate with a defined error (half width of the 95% confidence interval $\simeq 13\%$ of the estimated length for *C. suchus*, and between $\simeq 11\%$ for *M. leptorhynchus* and $\simeq 18\%$ for *G. gangeticus*, also without including *C. palustris*), while traditional methods are based mostly on subjective estimates during on-ground visual evaluations. We evaluated the variance structure of the different sources of imprecision for all 17 species (Table 1, Figures 5b and S2b–S17b). As an example, for *C. suchus* the variation distribution is fairly constant among individuals (Figure 5b) as follows: (i) head inclination, mean = 2.7% of the total variation; (ii) HL measurement errors (i.e., observer and hardware/software effects), mean = 46.8% of the total variation; (iii) natural allometric variability, mean = 47.8% of the total variation; (iv) allometry estimation (due to the number of observations from which the allometry coefficients are estimated), mean = 2.7% of the total variation (see Table 1 for the other species). The largest contribution to the imprecision came from natural allometric variability between individuals (i.e., within the species), which obviously cannot be reduced. The second largest contributor to imprecision is the accuracy of measurements from the photos, which in our case was limited by the camera resolution and, generally, had very little effect on the overall TL estimation (see above). This source of measurement imprecision, in addition to benefiting from a margin for improvement, is not constrained by the experience of the observer, which itself varies among species [47,48].

We then wanted to compare our framework's estimates to actual crocodiles from natural populations. Ideally, we would have taken precise measurements by hand from captured individuals and then taken pictures of the same individuals with the drone. Unfortunately, due to COVID travel limitations, we were unable to access wild or captive crocodiles. Consequently, we tested our reference allometric framework with orthorectified drone photos of *C. suchus* previously collected in 2018 in Niger and Benin [10]. We measured and estimated both $TL_p$ and $TL_e$, i.e., HL and TL measured directly from the drone photos ($TL_p$), and estimated TL estimated from our framework ($TL_e$) from $TL_p$. Most of the $TL_p$ fell within the CI of our $TL_e$ (n = 78 of 99), confirming the robustness of our predictions (Figure 6). The 21 $TL_p$ outside of the $TL_e$ CI were below the lower CI, which might suggest a tendency of the $TL_p$ method to underestimate the true size. The largest discrepancy between $TL_p$ and $TL_e$ was only 17 cm, which is biologically negligible for demographic classification of most individuals detectable by drones, because most crocodiles detected this way are typically >1.5 m TL (see below and Figure 7). Thus, the magnitude of any underestimation likely has little to no bearing for management.

Any discrepancy is likely mostly explained by measurement errors resulting from measuring a crocodile that is not lying perfectly straight. It can also be difficult to clearly identify the exact end of the crocodile tail on the photos and/or parts of the tail are missing or deformed due to past injuries [78], which will be difficult to see in drone photos. These same uncertainties probably affected the TL measurements in previous drone studies [33,34]. Ultimately, only on-ground TL measurements compared to $TL_e$ from drone-captured pictures of the same individuals will definitively confirm the robustness of our method. Regardless, our results already provide confidence in the framework, as most individuals fell within the CI (Figure 6). As a result, the size class distribution of a large sample would be only marginally affected and, with little to no bearing for population management or other demographic inferences, and access to demographic information from a greater portion of the population is worth this small trade-off.

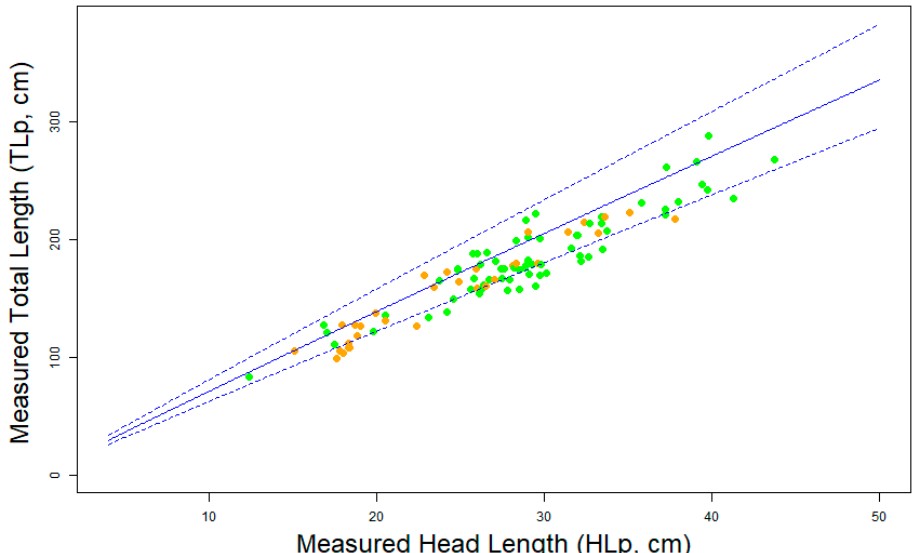

**Figure 6.** Assessing the accuracy of our reference allometric framework model to estimate the TL from HL measured in drone photos of crocodiles in natural populations. We measured head length (HL$_p$) and total length (TL$_p$) for *Crocodylus suchus* individuals detected in drone photos (from [10]) from the Tapoa River (W National Park, Niger) (light green dots, n = 67) and Bali Pond (Pendjari National Park, Benin) (orange dots, n = 32). For each HL$_p$ value, we also estimated the TL (TL$_e$) using our reference allometric framework, which is represented by the blue solid-line and including its 95% confidence interval envelope (blue dotted-lines). TL$_p$ were slightly lower than TL$_e$ in most cases, though not statistically significantly, and for 26 out of 99 individuals the TL$_p$ was below the 95% CI of TL$_e$.

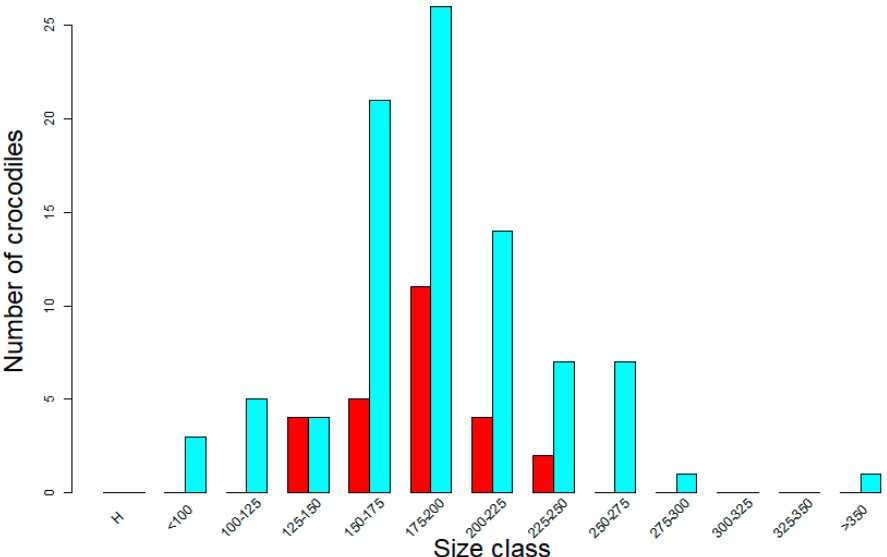

**Figure 7.** Size–class distribution inferred from drone-captured pictures. We counted *C. suchus* from pictures captured by drones in a 2018 survey in a 2 km-long transect of the Tapoa River, W National Park (Niger; n = 131; light blue bars) and the Bali pond (1.32 ha), Pendjari National Park (Benin; n = 38; red bars) [10]. Their total body length (TL$_e$) was estimated using our reference allometric framework, and they were then categorized as hatchlings (<50 cm), individuals smaller than 100 cm (<100), and then into 25 cm classes from TL = 100 to 350 cm, and over >350 cm. Some individuals were detected in the pictures but their HL could not be measured (blurred, partially visible, etc.) (Niger = 42, Benin = 12).

### 3.3. Improved Demographic Classification of Wild Crocodile Populations from Drones—But with Limitations

Using [10] data from 2018, we measured HL for *C. suchus* individuals along a 2 km transect of the Tapoa River (W National Park, Niger) and in the Bali Pond (Pendjari National Park, Benin). In Niger, we detected 226 individuals, including 25 individuals for which only the head was measurable (11% of the detected individuals) and 67 individuals for which both the head and full body length were measurable (30% of the detected crocodiles). In Benin, we detected 253 individuals, including 64 individuals for which only the head was measurable (25% of the detected individuals) and 32 individuals for which both the HL and TL were measurable (13% of the detected crocodiles). At both sites, neither the HL nor the TL were fully visible/measurable for the remaining $\pm$ 60% of individuals. Compared to previous drone-based approaches that measured only fully visible individuals, our method allowed us to capture usable demographic data for an additional 37% (Niger) to 200% (Benin) of the individuals within the detected sample, resulting in a more representative view of the population size-class distribution than was previously possible (e.g., as in the work presented in [34] or [33]).

We used the results of the best replicates (for which the number of detected crocodiles were the highest) of each transect (from the work presented in [10]) to obtain size-class distributions for the study areas in Niger and Benin. Using our reference allometric framework, we assigned each detected crocodile to a 25 cm $TL_e$ size class and obtained a fairly robust estimation of the population size-structure for individuals greater than 1.5 m TL, with a median in the 1.75–2.0 m size class (Figure 7). In Niger, only two individuals were larger than 2.75 m, with the largest estimated to be 3.97 m (First quartile: $Q_{inf}$ = 349.19 cm; Mean = 397.17 cm; third quartile: $Q_{sup}$ = 451.43 cm), which would be a very large contemporary individual for this species (M.H. Shirley, pers. obs.). A previous nocturnal spotlight survey in this area observed multiple large individuals (i.e., 2.5–3.0 m TL), as well as a balanced size class distribution including more than 46% of individuals juveniles (i.e., <1.5 m TL) [79,80].

As has been previously documented [10,34], drone surveys almost completely miss the small individuals (approximately less than 100 cm). They are often hidden in vegetation or are simply too small to detect or reliably identify even in very high resolution drone photos [10]. They are also predominantly nocturnal compared to adults, mostly to avoid predation risk from diurnal predators (e.g., birds, fish, mammals, snakes, and bigger crocodiles), which is less of a risk with increasing crocodile size [81,82]. For the same reasons, small crocodylians can also be difficult to detect using traditional survey methods [17]. However, for most crocodylian populations, hatchlings and juveniles represent more than 50% of the individuals and only a fraction of them will survive until adulthood [20,83]. A high proportion of juvenile size classes can be a good indicator of healthy populations because it represents high female fecundity, high juvenile survival, and high recruitment potential [84–87]. Thus, our inability to fully describe the size class distribution of crocodile populations remains one of the most significant limitations of drone-based approaches (but it also affects the more traditional methods).

## 4. Conclusions and Future Directions

Determining the size distribution of crocodylians in a population is critical to understanding population dynamics, designing and implementing management and conservation plans, and monitoring recovery [18,38]. Traditional, standardized diurnal and nocturnal population survey protocols rely on subjective, potentially biased, classifications from direct observations from a distance [47,49,88,89] or require capturing animals for direct measurements, which is costly, logistically challenging, requires experience, and comes with risks and stress to both researchers and the animals [61]. Drone-based methods can robustly estimate the total length of crocodylians without catching them and should be considered a viable approach by researchers and wildlife managers working on these species. Here we provide a fast, cheap, non-invasive, safe, and robust method that pre-

dicts crocodylian total length from head length measurements in drone photos with tight confidence intervals for 17 crocodylian species (Tables S1–S17). Moreover, the photos can be reanalyzed if necessary, making the method repeatable. Beyond use with drone photos, both our simple ratios and allometry results will be useful for researchers visually estimating TL from observations of HL during traditional surveys. Despite estimating the HL:TL allometric relationship for 17 different species, we recognize that drones are currently not very useful tools for surveying forest-dwelling crocodylian species (e.g., *Osteolaemus* spp., *Paleosuchus* spp., and *Mecistops* spp. in most habitats). We nonetheless provide the reference allometric framework for these species for the future with improvements in drone technology, for use with other methodologies, and/or in the event they are useful for other purposes.

Future improvements in drone technology will further improve crocodylian drone surveys and the data we are able to capture remotely. Even recent technological developments have extended drone flight ranges, and new hybrid drones use less energy during flights and considerably increase their autonomy [90]. Using more powerful, on-board optical equipment can increase image and video resolution, reducing the current overestimation bias when estimating total length. Though crocodylians are heterothermic and expected to maintain body temperatures closely matching ambient, drone-mounted infrared cameras may eventually result in increased detection of all size classes during nocturnal flights, especially in the early hours of the evening when the ambient drops quickly but crocodylians are still warm from the day [91]. Drone positioning, and thus, Ground-Sample Distance (GSD) estimation, could be significantly improved by using onboard Real Time Kinematic (RTK) or Post Processed Kinematic (PPK) GPS correction technology, and by designing ground control points (GCPs) for orthophotographic correction [92]. However, these technologies will considerably increase the logistical costs and the required technical skills for a limited and sometimes unnecessary gain in precision. Another significant improvement could be provided through automation—counting and measuring crocodiles on map images is a tedious, time-consuming task that requires intense concentration. The characteristic triangular shape of the crocodylian head on aerial images may allow for the automation of counting and potentially measuring individuals using trained AI models, which are currently under development for other species [93].

Among their many current applications, drones can be used to remotely identify specific individuals from dorsal scute patterns [32]. And, in the future may help facilitate the management of human-crocodile conflicts [94]. The ability to quickly detect, identify, and measure large crocodiles will be advantageous in areas of high conflict, including identifying sites at risk [95]. Drones have even been used to remotely capture crocodiles, which will be useful in the case of problem animals that are often wary, trap shy, or otherwise unapproachable for capture using traditional methods [96]. As the technology embedded improves and their prices drop, drones will be an increasingly accessible application in conservation and environmental management, even in impoverished areas.

**Supplementary Materials:** The following supporting information can be downloaded at: https://www.mdpi.com/article/10.3390/drones8030115/s1. Figure S1: A priori distribution of crocodiles head inclination and Gaussian vs. Johnson's distribution; Figure S2: Allometric relationship between head length (HL) and total length (TL) in log-log scale of wild-caught American alligator (*Alligator mississippiensis*) measured in natural populations; Figure S3: Allometric relationship between head length (HL) and total length (TL) in log-log scale of wild-caught spectacled caiman (*Caiman crocodilus*) measured in natural populations (n = 459); Figure S4: Allometric relationship between head length (HL) and total length (TL) in log-log scale of wild-caught American crocodile (*Crocodylus acutus*) measured in natural populations (n = 906); Figure S5: Allometric relationship between head length (HL) and total length (TL) in log-log scale of wild-caught Orinoco crocodile (*Crocodylus intermedius*) measured in natural populations (n = 403); Figure S6: Allometric relationship between head length (HL) and total length (TL) in log-log scale of wild-caught freshwater crocodile (*Crocodylus johnstoni*) measured in natural populations (n = 588); Figure S7: Allometric relationship between head length (HL) and total length (TL) in log-log scale of wild-caught Morelet's crocodile (*Crocodylus moreletii*)

measured in natural populations (n = 597); Figure S8: Allometric relationship between head length (HL) and total length (TL) in log-log scale of wild-caught Nile crocodile (*Crocodylus niloticus*) measured in natural populations (n = 340); Figure S9: Allometric relationship between head length (HL) and total length (TL) in log-log scale of wild-caught mugger crocodile (*Crocodylus palustris*) measured in natural populations (n = 80); Figure S10: Allometric relationship between head length (HL) and total length (TL) in log-log scale of wild-caught saltwater crocodile (*Crocodylus porosus*) measured in natural populations (n = 370); Figure S11: Allometric relationship between head length (HL) and total length (TL) in log-log scale of wild-caught Cuban crocodile (*Crocodylus rhombifer*) measured in natural populations (n = 196); Figure S12: Allometric relationship between head length (HL) and total length (TL) in log-log scale of wild-caught gharial (*Gavialis gangeticus*) measured in natural populations (n = 353); Figure S13: Allometric relationship between head length (HL) and total length (TL) in log-log scale of wild-caught Central African slender-snouted crocodile (*Mecistops leptorhynchus*) measured in natural populations (n = 159); Figure S14: Allometric relationship between head length (HL) and total length (TL) in log-log scale of wild-caught black caiman (*Melanosuchus niger*) measured in natural populations (n = 167); Figure S15: Allometric relationship between head length (HL) and total length (TL) in log-log scale of wild-caught dwarf crocodile (*Osteolaemus tetraspis*) measured in natural populations (n = 106); Figure S16: Allometric relationship between head length (HL) and total length (TL) in log-log scale of wild-caught Cuvier's dwarf caiman (*Paleosuchus palpebrosus*) measured in natural populations (n = 149); Figure S17: Allometric relationship between head length (HL) and total length (TL) in log-log scale of wild-caught smooth-fronted caiman (*Paleosuchus trigonatus*) measured in natural populations (n = 87). Table S1: *Alligator mississippiensis* framework. Table S2: *Caiman crocodilus* framework; Table S3: *Crocodylus acutus* framework; Table S4: *Crocodylus intermedius* framework; Table S5: *Crocodylus johnstoni* framework; Table S6: *Crocodylus moreletii* framework; Table S7: *Crocodylus niloticus* framework; Table S8: *Crocodylus palustris* framework; Table S9: *Crocodylus porosus* framework; Table S10: *Crocodylus rhombifer* framework; Table S11: *Crocodylus suchus* framework; Table S12: *Gavialis gangeticus* framework; Table S13: *Mecistops leptorhynchus* framework; Table S14: *Melanosuchus niger* framework; Table S15: *Osteolaemus tetraspis* framework; Table S16: *Paleosuchus palpebrosus* framework; Table S17: *Paleosuchus trigonatus* framework.

**Author Contributions:** Conceptualization, Investigation, Methodology, Project Administration, Resources, Validation and Writing—Original Draft: C.A., G.L.M. and M.H.S.; Formal analysis: C.A. and G.L.M.; Supervision: G.L.M. and M.H.S.; Data Curation, Funding Acquisition, Investigation, Writing—Editing and Review: C.A., G.L.M., A.V., X.C., J.W.L., P.G., G.P.-S., E.P., P.C., F.V., I.J.R., B.M., J.E.C., A.M., A.R.W., R.S., M.T., M.B. and M.H.S. All authors have read and agreed to the published version of the manuscript.

**Funding:** This project was funded by the Fondation BIOTOPE, IUCN/SSC Crocodile Specialist Group Student Research Assistance Scheme, European Croc Networking Meeting (ECNM) grants, Fonds de Solidarité ét de Développement des Initiatives Étudiantes de l'Université de Montpellier, Ondulia, an Okpal crowdfunding campaign, and Projet Mecistops. Fieldwork by MH Shirley was funded by the Wildlife Conservation Society, WCS's Wildlife, Landscapes and Development for Conservation in Northern Uganda project (WILDCO) supported by USAID, a National Science Foundation Doctoral Dissertation Improvement Grant (DDIG; Agreement No. 1010574), the Conservation, Food, and Health Foundation, Riverbanks Zoo and Gardens Conservation Support Fund, U.S. Fish and Wildlife Service Wildlife Without Borders Program (Agreement No. 96200-1-G003), Cleveland Metroparks Zoo, Oklahoma City Zoo, Aspinall Foundation, WWF, Columbus Zoological Park Association, Inc. Conservation Fund, IDEA WILD, St Augustine Alligator Farm, Rotary International, IUCN/SSC Crocodile Specialist Group, AZA Crocodilian Advisory Group, Minnesota Zoo, Fresno Chaffee Zoo, San Diego Zoological Society, Mohamed bin Zayed Species Conservation Fund, C. Stevenson, and JP Ross. The Rare Species Conservatory Foundation served as a zero-overhead funding fiduciary for grants. Sampling and animal handling methods were reviewed and approved by IFAS ARC at the University of Florida (approval # 011-09WEC). We thank the Wildlife Conservation Society (WCS) through Instituto Piagaçu (IPI), Instituto de Desenvolvimento Sustentável Mamirauá (IDSM) and the National Council for Scientific and Technological Development (CNPq) which provided financial support. CNPq also provided doctoral scholarship for Boris Marioni. Igor Joventino Roberto thanks Coordenação de Aperfeiçoamento de Pessoal de Nível Superior—Brasil (CAPES), the CNPq for fellowship (PDCTR 301304/2022-0) and Crocodylia Brasil for the support. We thank WCS-Ecuador which provided financial support to Francisco Villamarín.

**Data Availability Statement:** The data presented in this study are available on request at the following DOI: https://doi.org/10.57745/LEVTCU.

**Acknowledgments:** We thank Jean-Loup, Françoise, and Martine Claret for their permission to access the experimental flight area, and Pierrick Labbé for his help. We thank Nature Conserv'Action (NCA) for providing the drones. We thank the Zoological Society of London (ZSL), the staff of Centre National de Gestion des Réserves de Faune of Benin (CENAGREF), and the Ministère des Eaux et Forêts of Niger for authorizing the research that provided drone images for *C. suchus*. In Benin, this research was authorized under permit number 110/17/CENAGREF/DG/DT/DAF/AD from CENAGREF. We would like to thank W. Oduro, H. Yaokokore-Beibro, C. Ofori-Boateng, C. Hutton, M. Jallow, K. Ingenloff, L. Paziaud, M. Selinske, Z. Chifundera, S. Aucoin, the staff of Kidepo Valley National Park, Wildlife Conservation Society Uganda, and all other persons at the protected areas and other sites where we captured *C. suchus* to take measurements for their help with fieldwork, logistics, permits, and sampling. Logistical support in Gabon and DR Congo was provided by the Wildlife Conservation Society, World Wildlife Fund (WWF), Smithsonian Institute, Project Protections des Gorilles (PPG) and the Aspinall Foundation, Protection des Grandes Singes de Moukalaba-Doudou (PROGRAM), Fondation Liambissi and the Lukuru Foundation. We thank R. Starkey, M. Starkey, R. Calaque, B. Huijbrechts, B. Verhage, L. Korte, M. Butler, N. Bout, A. Vosper, K. Kombila, R. Beville, F. Koumba Pambo, J. and T. Hart, P. and S. du Plessis for their assistance throughout. We thank M. Adu-Nsiah of the Ghana Wildlife Division (Ghana), Col. K. Amani Denis of the Direction de la Faune, Ministère des Eaux et Forets (Côte-d'Ivoire), A. Jallow of the Dept. of Parks and Wildlife Management (Gambia), and P. Anying and A. Rwetsiba of the Uganda Wildlife Authority (Uganda) for authorizing the research that provided the body measurements for *C. suchus*. In Gabon and DR Congo, we thank the Centre National pour la Recherche Scientifique et Technologique (CENAREST—N°AR0024/09/ and N°AR0013/11) for authorising research in the Gabonese national territory. We thank the Agence Nationale des Parcs Nationaux for authorising research in Loango National Park, and the support of former conservateur Brice Leandre Meye and Flore Aurelie Koumba Pambo. Finally, we would like to thank all the peoples and organizations involved in the collection of crocodilians measurement data used on this paper. Namely Omar Hernández, Lilo Enes, Roldan De Sola, Arnaldo Ferrer, Jonathan Triminio. We acknowledge and thank Yank Moore for significant contributions involved in capture, handling, and data collections of *A. mississippiensis* on Jekyll Island. We thank the Cuban staff, led by Etiam Perez and Gustavo Sosa, for providing the *C. rhombifer* information collected at the Zapata Swamp. We thank Colleen Downs and Jon Warner from the University of KwaZulu-Natal's Zululand Crocodile Research Program, School of Life Sciences, University of KwaZulu-Natal, South Africa and James Hennessy from the The National Reptile Zoo, Ireland.

**Conflicts of Interest:** The authors declare no conflicts of interest.

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
