# Peer review of "Estimating Total Length of Partially Submerged Crocodylians from Drone Imagery"

_drones, doi:10.3390/drones8030115_

Round 1
Reviewer 1 Report
Comments and Suggestions for Authors
The authors present a novel method for estimating body length of crocodylians from drone imagery that has the potential to better inform management and conservation of these cryptic species. In particular, the authors should be commended for their thorough description of this methodology, the analysis used to verify its accuracy for a variety of species and imagery, and discussion of the limitations of the method. All this contributes to ensuring the method can be reproduce and used to full effect by researchers and conservation managers under wide range of circumstances. I therefore recommend this article be published with minor revisions, which I have outlined for the authors in the 'specific comments' section of the attached document.

Use of English language is appropriate for an academic article and easy to understand.
Reviewer 2 Report
Comments and Suggestions for Authors
The manuscript showed interesting results about crocodile’s allometry research and images obtained by UAVs; it is innovative and presents original results. I found minor errors throughout the manuscript; however some of them must be reviewed. I attached a review detail.
